# The Allosteric Antagonist of the Sigma-2 Receptors—Elayta (CT1812) as a Therapeutic Candidate for Mild to Moderate Alzheimer’s Disease: A Scoping Systematic Review

**DOI:** 10.3390/life13010001

**Published:** 2022-12-20

**Authors:** Anum Rasheed, Ahmad Bin Zaheer, Aqsa Munawwar, Zouina Sarfraz, Azza Sarfraz, Karla Robles-Velasco, Ivan Cherrez-Ojeda

**Affiliations:** 1Department of Research, Services Institute of Medical Sciences, Lahore 54000, Pakistan; 2Department of Research, Al Nafees Medical College and Hospital, Isra University, Islamabad 44000, Pakistan; 3Department of Research and Publications, Fatima Jinnah Medical University, Lahore 54000, Pakistan; 4Department of Pediatrics and Child Health, The Aga Khan University, Karachi 74000, Pakistan; 5Department of Allergy, Immunology & Pulmonary Medicine, Universidad Espíritu Santo, Samborondón 092301, Ecuador

**Keywords:** CT1812, Elayta, sigma-2, therapeutic candidate, Alzheimer’s disease, clinical trials, quality of life, public health

## Abstract

Nearly 35 million people worldwide live with Alzheimer’s disease (AD). The prevalence of the disease is expected to rise two-fold by 2050. With only symptomatic treatment options available, it is essential to understand the developments and existing evidence that aims to target brain pathology and dementia outcomes. This scoping systematic review aimed to collate existing evidence of CT1812 for use in patients with AD and summarize the methodologies of ongoing trials. Adhering to PRISMA Statement 2020 guidelines, PubMed/MEDLINE, Embase, Cochrane, and ClinicalTrials.gov were systematically searched through up to 15 November 2022 by applying the following keywords: CT1812, Alzheimer’s disease, dementia, and/or sigma-2 receptor. Three completed clinical trials were included along with three ongoing records of clinical trials. The three completed trials were in Phases I and II of testing. The sample size across all three trials was 135. CT1812 reached endpoints across the trials and obtained a maximum concentration in the cerebrospinal fluid with 97–98% receptor occupancy. The findings of this systematic review must be used with caution as the results, while mostly favorable so far, must be replicated in higher-powered, placebo-controlled Phase II–III trials.

## 1. Introduction

Around 6.2 million people in the United States (US) have Alzheimer’s disease (AD); there are 35 million people with AD worldwide [1,2]. The prevalence is expected to increase two-fold by the year 2050 [3,4,5]. AD’s cost of healthcare is projected to increase beyond $300 billion in the US alone [6,7]. With worldwide costs estimated to reach $1 trillion by 2050, over half of the afflicted population has mild AD (50.4%), whereas 30.3% have moderate AD and 19.3% have severe AD [8,9,10,11,12]. CT1812 also known as Elayta is a small-molecule antagonist of the sigma-2 receptor; it binds to the receptors at the progesterone receptor membrane component 1 subunit [13,14]. The mechanism of action is as follows [15,16]: CT812 penetrates the blood–brain barrier and selectively binds to the sigma-2 receptor. CT1812 is a high brain penetrant that works by displacing Ab oligomers that are bound to neuronal receptors at the synapses [17]. As a lipophilic isoindoline, CT1812 is formed as a fumarate salt that works similarly to the related class of compounds, which have high specificity and affinity for the sigma-2 receptors; this is a central regulator of oligomer receptors [17,18,19]. The binding of sigma-2 receptors destabilizes the Ab oligomers’ binding site, leading to increases in the off-rate of the oligomers, and these are then cleared into the cerebrospinal fluid [17,18,19]. Pre-clinical models have found that the drugs in the CT family occupy or exceed 80% prevention of downstream synaptotoxicity and restore memory in aged transgenic mouse models of Alzheimer’s disease [17,18,19].

Disease-modifying agents for degenerative disorders including AD are scientifically backed by decades of testing of the biology of the synaptic function and plasticity in pre-clinical and clinical settings [20,21,22]. CT1812 is intended to target age-related degenerative disorders of the central nervous system and the retina such as AD, dementia with Lewy bodies, Parkinson’s disease, and dry age-related macular degeneration [23,24,25]. Medication is currently being tested in multiple Phase II trials, with preclinical Phase I testing being conducted, and Phase III studies underway. In 2021, three studies were completed named SNAP, SHINE (cohort 1), and SPARC to evaluate the safety and tolerability of CT1812. In 2022, PK, SEQUEL, Human AME, and PK were key studies of intertest, whereas, for 2023 and beyond, studies including Phase II trials in DLB and SHINE (cohort 2), Phase II with ACTC, and Phase II in dry AMD are ongoing [26]. They will be further elucidated in the results and discussion sections. CT1812 is currently an investigational therapy that has not been approved by the U.S. Food and Drug Administration (FDA); post the conduction of adequate placebo-controlled clinical trials, the risk–benefit profile will be conducted [27]. However, Elayta has been granted fast-track designation by the U.S. FDA which promotes the development and expediting of the agency’s review of new compounds in treating serious conditions with an unmet medical need [27].

The chemical structure of CT1812 is depicted in Figure 1. It is also known as Sigma-2 receptor antagonist 1 and CT1812 [15,28]. As per the ICD-11 Code, it is currently utilized for AD (8A20) and is a novel approach to AD modification [28]. The role of CT1812 for AD has insofar been neuroprotective and current evidence suggests that it reduces cognitive deficits and neuroinflammation [29]. 

With AD being the most common cause of dementia worldwide, the treatment is only targeted at symptomatic therapy. With an abundance of clinical trial data underway [30], it is necessary to explore the recent developments in treatment that are targeted at the overall burden of AD and pathology within the brain. Clinical research is advancing toward more definitive therapeutic candidates to treat the hallmark pathologies in AD [29,31,32]. This scoping systematic review will summarize existing evidence and also describe future trials with CT1812 among patients with AD.

## 2. Materials and Methods

PubMed/MEDLINE, Embase, Cochrane Library, and ClinicalTrials.gov were systematically searched for primarily clinical studies adhering to PRISMA Statement 2020 guidelines [33]. The following keywords were used across the databases and clinical trial register up until November 15, 2022: “CT1812, Alzheimer’s disease, dementia and/or sigma-2 receptor”. The Boolean logic (and/or) was applied during the search phase. No language restrictions were applied; in cases where the study was not in English, it was translated into English using Google Translate. 

The inclusion criteria comprised primarily clinical studies with adult human patients aged 18 or above, of any gender, diagnosed with Alzheimer’s disease and being intervened with CT1812 in a controlled setting. The participants were required to be intervened with CT1812 and to have a placebo group. Clinical trial records were included to discuss the ongoing clinical trials of CT1812 for AD. Case series/reports, systematic reviews, meta-analytical studies, and letters to editors were excluded. 

During the screening phase, the titles and abstracts of the shortlisted studies from the databases and the titles and summaries of clinical trial records were reviewed independently by two reviewers (Z.S. and A.S.). A third reviewer was present to resolve any disagreements (I.C.O.). The PRISMA flowchart depicting the study selection process is attached in Figure 2. The bibliographic entries were stored in EndNote X9 (Clarivate, London, UK), where the duplicates were removed during the study selection process. A kappa score was computed to assess the inter-rater reliability, which measured the level of agreement between the two reviewers in the Statistical Package for Social Sciences (SPSS, v25). 

The data were collected onto a shared spreadsheet. Firstly, they were reported as characteristics of the included trials (trial ID, author, year, country, title, journal, phase and design of the study, inclusion criteria, pharmacologic intervention, and outcome measures). Secondly, a tabulation of the sample size, age in years, gender, efficacy, and safety was presented. Lastly, for the ongoing trials, the NCT number, title, current status, conditions being studied, interventions, outcome measures, participants, phases, enrollment, study type, design identifiers, primary completion date, and locations were listed.

The included randomized trials were qualitatively assessed for the risk of bias using version 2 of the Cochrane risk-of-bias tool for randomized trials (RoB 2) [34]. The risk of bias assessment was assessed independently by two reviewers (Z.S. and A.S.), and a final consensus was reached. The RoB 2 tool comprises five domains: (1) bias in the randomization process, (2) bias in deviating from the intended interventions, (3) bias with missing outcome data, (4) bias in measuring the outcome, and (5) bias in selecting the reported results. The summary finding plot was generated where domain-level judgments were reported as (1) low risk of bias, (2) some concerns, and (3) high risk of bias. These were reported as a traffic light plot of bias assessment across all individual studies.

## 3. Results

A total of 237 study records and 9 clinical trial records were identified (n = 246). Of these, 18 duplicate records were removed. During the screening phase, 228 records were screened for titles and abstracts/summaries. Of these, 215 were excluded as they did not warrant inclusion. Therefore, 13 studies were assessed for eligibility. Of these, seven were removed as four were non-human studies and three were either reviews or letters. Finally, three studies and three records were included in this study. The kappa score of the inter-reviewer agreement was 0.914, suggesting excellent agreement among the reviewers.

### 3.1. Characteristics of the Included Trials

The characteristics of the included trials are listed in Table 1.

Izzo et al. (2021) conducted a Phase Ib/IIa clinical trial in Australia and enrolled 19 participants with mild to moderate Alzheimer’s disease. The participants were randomized to either Elayta (90, 280, or 560 mg) or a placebo for 28 days. The safety/efficacy and pharmacokinetics were the primary outcomes of the study and the changes in protein biomarkers and cognitive outcomes were explored.

Hamby (2020) conducted a Phase I/II randomized, double-blind placebo-controlled parallel group study in the US. The participants were administered CT1812 at 300 mg, 100 mg dosages, or placebo for up to 180 days. The outcomes of treatment-related adverse events and serious adverse events over 30 weeks were assessed. In addition, changes in the brain synaptic density over the six months using SV2A PET ligand 11C-UCB-J were reported.

Grundman et al. (2019) conducted a Phase II clinical trial in Melbourne, Australia, among individuals aged 18–55 or 65–75. The trial was conducted in two parts: A single ascending dose (SAD)/food-effect study (part A) and a multiple ascending dose (MAD) study (part B). Safety, tolerability, plasma pharmacokinetics, and drug concentrations in the cerebrospinal fluid were assessed. 

### 3.2. Participant Characteristics, Safety, and Efficacy Outcomes

The patient characteristics, safety, and efficacy outcomes are depicted in Table 2. 

Izzo et al. (2021) enrolled a total of 19 participants of whom 9 were female. The mean age was 70.2 years (SD = 9.2). On noting efficacy outcomes, the concentration of Aβ oligomers was measured via Western blot analysis in CT1812-treated patients. The CSF increases in Aβ oligomers were compared to the patient’s baseline and with the placebo using the student’s *t*-test (*p* = 0.014). The findings supported evidence of clinical target engagement. In addition, neurogranin B (via ELISA) and synaptotagmin-1C (via LC-MS/MS) concentration have measured the covariance that was tested compared to the patients’ baseline and the with placebo. The former (*p* = 0.05) and the latter (*p* = 0.011) both decreased in concentration.

Hamby (2020) enrolled 23 participants of whom 11 were female. The mean age was 70 years (SD = 8.8). Concerning efficacy, the values were recorded until day 180, and findings were reported as mean score change from baseline and (standard error, SE). The distribution volume ratio (DVR) change from baseline for 300 mg, 100 mg, and placebo groups was −0.043 (0.02), −0.019 (0.02), and 0 (0.02), respectively. Concerning Aβ oligomers, the mean score changes from baseline over six months for 300 mg, 100 mg, and placebo groups were −594 (381.57), 517.7 (445.24), and 222.25 (397.27), respectively, whereas, for Tau, the six-month mean score change outcomes for 300 mg, 100 mg, and placebo groups were −84.08 (111.96), 121.22 (131.96), and 36.74 (120.7), respectively. The cognitive composite score mean changes for 300 mg, 100 mg, and placebo groups were −0.33 (0.109), −0.11 (0.102), and −0.1 (0.109), respectively. For MMSE mean score change outcomes in 300 mg, 100 mg, and placebo groups the findings were −2.92 (1.54), −1.26 (1.45), and −0.74 (1.55), respectively.

Grundman et al. (2019) enrolled a total of 93 participants with a total of 27 females. The median age ranged from 26–69 and the interquartile ranges across the subgroups were 19 to 73 years. Efficacy outcomes were obtained for CT1812 CSF concentrations, which increased in a dose-dependent manner across two orders of magnitude in a two-week intervention period. With concentrations of 560–840 mg, the average CSF concentration reached 97–98% receptor occupancy. However, given the 2-week intervention duration, the cognitive scores were similar across the elderly cohort before and after treatment.

On noting safety outcomes, Izzo et al. reported no adverse events concerning the safety of the participants, whereas Hamby reported mild treatment-emergent adverse events among 85.7% of intervened patients (high dose-300 mg). In comparison, 50% of low-dose (100 mg) patients had mild adverse events and 66.7% of patients receiving a placebo had mild adverse events. There were no major differences in the discontinuation of the trial among the CT1812 and placebo groups, with 26.7% and 16.7% trends, respectively. Finally, Grundman et al. reported that 43% of patients had adverse events post-CT1812 intervention and 17% reported adverse events after the placebo. On the whole, no deaths or serious adverse events were reported.

### 3.3. Ongoing Clinical Trials

Three ongoing clinical trials were located in the Clinical Trials Registry [38]. They are tabulated in Table 3.

NCT05531656 also referred to as COG0203 is expected to enroll 540 participants using a randomized parallel-group, quadruple masking curative Phase II design. Males and females aged between 50–85 years with early AD are to be enrolled. The key outcomes include changes from baseline in CDR-SB (clinical dementia rating scale sum of boxes), ADAS-Cog 13 (Alzheimer’s disease assessment scale–cognition), ADCS-ADL (activities of daily living scale), CSF (cerebrospinal fluid) concentrations of drug concentration and plasma measures of changes, and volumetric MRI outcomes including hippocampal and whole brain volume change. The trial’s primary completion date is August 2026. 

NCT03507790 also referred to as COG0201 is enrolling 144 participants with a randomized, parallel assignment, quadruple masking and curative Phase II design. Both males and females aged 50 to 85 years with mild to moderate AD have been enrolled. The primary outcome measure includes the number of participants with treatment-related adverse events and serious adverse events. The trial is being conducted in the US and Australia and is expected to reach completion by 24 September 2023.

NCT04735536 also known as COG0202 is set to enroll 16 participants using a randomized, crossover assignment with quadruple masking and curative modality with a Phase II design. The trial is enrolling participants until 30 March 2023 and is being conducted in Amsterdam, Netherlands. The listed primary outcome is the measurement of the CT1812 plasma concentration ratio. The participants are both males and females aged 50 to 85 years. 

### 3.4. Quality Appraisal of the Included Trials

For bias arising due to randomization, two trials had low concerns, whereas one trial had some concerns. All three trials had concerns about bias arising due to deviations from the intended interventions. For bias arising due to missing outcome data, one trial had some concerns, whereas two trials had low concerns. On noting the bias in the measurement of the outcome, one trial had some concerns, whereas two trials had low concerns. For bias in the selection of the reported result, two trials had some concerns, whereas one trial had low concerns. Finally, the overall risk of bias was low across two trials and one trial had some concerns (Figure 3).

Izzo et al. (2021) presented no concerns during the process of randomization, whereas, Hamby (2020) had some concerns about the selection of the reported result. This was a concern as the trial’s primary outcome measures consisted of treatment-related adverse events and changes in brain synaptic density over six months. There were some concerns about the selection of the reported result, as ascertained by the two reviewers (Z.S. and A.S.). In comparison, Grundman et al. (2019) had some concerns about the randomization process. Since the trial was a Phase I trial, the authors also included healthy and young subjects who received doses of CT1812 between 10 and 1120 mg. The concerns about missing outcome data and reporting were present since the serum concentrations of CT1812 were reported as primary outcomes whereas cognitive testing was performed on day 14 of treatment with no longer-term follow-up. The summary findings are depicted in Figure 3.

## 4. Discussion

In this scoping systematic review, we included a total of three completed clinical trials and a synthesis of three ongoing clinical trials was presented. The three completed clinical trials were in Phase I/II. Two of them were conducted in Australia whereas one was conducted in the US. CT1812 was administered in different dosages ranging from 10 to 1120 mg across the differing arms. The total sample size of all three trials was 135 participants. The safety and efficacy outcomes were presented whereby comparable safety findings were noticed for the intervention and placebo groups. CT1812 reached given endpoints across the three trials with a maximum CSF concentration of 97–98% receptor occupancy.

A Phase I COG01013 trial enrolled 14 healthy individuals who received six daily oral doses of CT1812. Catalano shared the pre-AD test findings and stated that CT1812 had no food–drug interactions and the safety findings were strong [39]. The Phase I safety trial of the Aβ oligomer receptor antagonist CT1812 was first presented at the Alzheimer’s Association International Conference in 2017. Overall, there were no reported drug–drug interactions [39]. Catalano and colleagues conducted a single ascending dose part six cohorts of CT1812-treated (n = 6) and placebo-treated (n = 2) participants [40]. The notable findings were that CT1812 tested the oligomer hypothesis for AD treatment [40]. The authors stated that future development is key in testing the ability of the drug in improving cognitive ability in patients with AD [40].

In comparison, Grundman et al. (2019) in their Phase I trial reported CT1812 as generally safe and well tolerated; however, the adverse events consisted of mild headaches or gastrointestinal disturbances [17]. The CSF testing corroborated that the drug penetrated the human blood–brain barrier and that extrapolations from mice studies exceeded the expected minimum concentrations required to improve memory [17]. At a dosage of 560 mg, the CT1812 levels penetrated 97–98% of the receptor occupancy in transgenic mice, whereas, at the 840 mg dose, the CSF levels in humans acquired 98% receptor occupancy [17]. As a novel, small molecular that is a brain-penetrating antagonist preventing binding of neuronal receptors to AβOs [41], the findings are suggestive of the notion that CT1812 can allosterically inhibit both [17]. However, given that AD is a complex disease, the treatment approach will likely be multifactorial [42,43,44]. While the study assesses the role of Elayta as a novel therapeutic candidate for mild/moderate AD, it is imperative to acknowledge that the medication alone may be unable to completely ameliorate the negative effects of increased AβOs concentrations, which are contributors to the progression of the disease [17].

A preprint paper by Colom–Cadena in 2021 aimed to understand the knowledge gap towards the synaptic accumulation of Aβ and subsequent synapse degeneration [45]. The authors studied TMEM97, which is also a sigma intracellular receptor 2. The authors imaged over one million individual synaptic terminals in the temporal cortex from nine AD cases and six age-matched controls [45]. The study was successful in inhibiting the Aβ-TMEM97 interaction with the APP/PS1+Tau mouse model of AD; this was achieved by using CT1812 (n = 20); the drug concentration was negatively associated with synaptic Förster resonance energy transfer (FRET) signals between Aβ and TMEM97 [45]. This study implies that the synaptic binding of Aβ in the human brain with AD is where the synaptotoxicity can be mediated [46].

Earlier, sigma receptors were incorrectly classified as a subtype of opiate receptors [47]. With subsequent studies conducted in this area, the sigma receptors were classified as a distinct class of receptors as sigma-1 and sigma-2 [48,49]. The separate identification is useful as molecular imaging of sigma receptors may be conducted using noninvasive techniques including positron emission tomography (PET), which is used for monitoring the treatment effects of sigma-targeted therapeutic agents such as CT1812 [50]. To date, over 200 human mutations have been found to cause early-onset familial AD [51,52,53]. Of note, almost all of these mutations lead to a single phenotype, increased Aβ. The early-onset mutations either increase the total amount of Aβ or the relative proportion of the longer form of the protein. The therapeutic candidate CT1812 lowers Aβ oligomer binding affinity to neuronal synapses and has restored cognitive function in mice along with currently being studied in various Phase II clinical trials with AD patients [49].

### 4.1. Limitations

There are certain limitations to this study. First, while the highest quality of RCT evidence was utilized, the outcomes were heterogeneous across the trials. Second, given the paucity of the literature in this area, grey literature was cited in the discussion. Third, the findings are far from generalizable and only serve as a vital piece of information as AD research progresses toward a definitive therapeutic candidate to improve dementia outcomes. Fourth, a quantitative meta-analysis could not be conducted due to the diversity in reported outcomes. While the studies had either a low risk of bias or some concerns, Grundman et al.’s 2019 trial raised some concerns about randomization, missing outcome data, and selection of the reported results, while Hamby’s (2020) trial raised only some concerns in the selection of the reported result. Izzo and colleagues (2021) raised no concerns about the risk of bias in any domain. In an effort to ensure best practice, two reviewers independently assessed each included trial and reached a consensus about the final assessment. For this reason, the authors of this review are confident in the completeness of the evaluation of included studies for the risk of bias.

### 4.2. Implications for Clinical Care

Our review summarizes evidence from the existing literature in the area of improving clinical care for Alzheimer’s disease. As a candidate therapy, CT1812 is still under development for age-related degenerative diseases. With trials in this area currently ongoing, as listed in Table 3, we anticipate more robust and concrete evidence solidifying the use of CT1812 for patient populations. Notably, there are currently no approved therapies for dementia with Lewy bodies; this type of dementia impacts nearly 1.4 million people in the United States alone and is the second most common form of dementia. Individuals that present with motor deficits and cognitive and behavioral changes make it difficult to diagnose them with any specific type of dementia. With preliminary evidence pointing toward CT1812’s effectiveness in reversing deficits in in vitro models, various proof-of-concept studies have now progressed toward Phase II and III of testing. With such approaches, the care for an incurable disease, dementia, may achieve palpable progress. With prospective placebo-controlled trials, the authors of this study believe that elderly individuals with dementia due to Alzheimer’s disease may have expanded treatment options.

## 5. Conclusions

The prospect of CT1812 as an allosteric antagonist of sigma-2 receptors heightens the likelihood of finding a therapeutic candidate for mild to moderate Alzheimer’s disease. With the ongoing and published clinical trial data utilized in this scoping systematic review, there is cautious optimism toward the benefits that CT1812 can incur among AD patients with dementia. So far, there are noteworthy central nervous system effects with the treatment and the pre-clinical to clinical journey for Ab oligomers and sigma-2 receptors that require further exploration in Phase II and III high-powered, placebo-controlled trials.

## Figures and Tables

**Figure 1 life-13-00001-f001:**
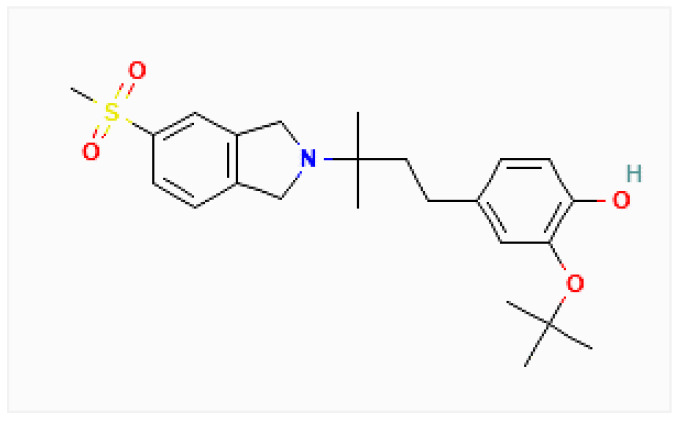
Chemical structure of CT1812.

**Figure 2 life-13-00001-f002:**
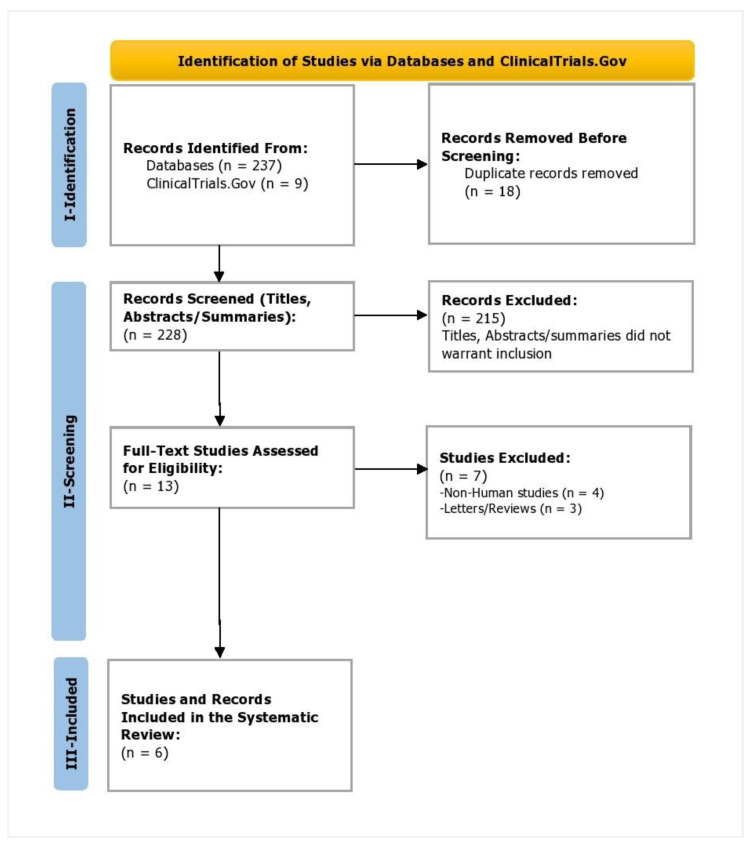
PRISMA flowchart depicting study selection.

**Figure 3 life-13-00001-f003:**
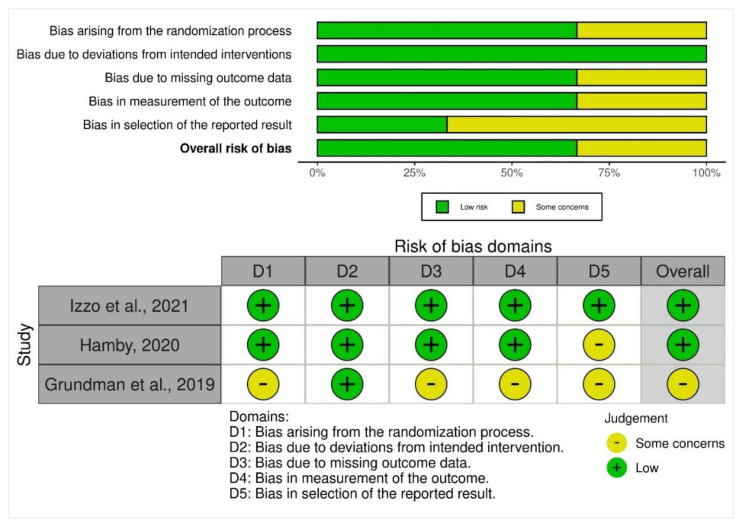
Rob2 tool summary plot findings and traffic light plot depicting individual study findings [17,29,36].

**Table 1 life-13-00001-t001:** Characteristics of the included trials.

Trial ID	Author	Year	Country	Title	Journal	Phase	Design	Inclusion Criteria	Pharmacologic Intervention	Outcome Measures
NCT02907567 [35]	Izzo et al.	2021	Australia	Preclinical and clinical biomarker studies of CT1812: A novel approach to Alzheimer’s disease modification	Alzheimer’s and Dementia	Phase Ib/IIa	Randomized, double-blind, placebo-controlled	Men and women aged 50–80 with mild to moderate AD as per the 2011 NIA-AA criteria, with MRI diagnosis, MMSE (18–26), without active depression, and a GDS < 6	Two doses of CT1812 in adults with mild to moderate AD for 28 days compared to a placebo	Treatment-emergent adverse events (physical exams; monitoring of vital signs, ECGs, and clinical and laboratory assessments)
NCT03493282 [36]	Hamby	2020	USA	A Pilot Synaptic Vesicle Glycoprotein 2A (SV2A) PET Study to Evaluate the Effect of CT1812 Treatment on Synaptic Density in Participants With Mild to Moderate Alzheimer’s Disease	Alzheimer’s Association International Conference	Phase I/II	Single-center, randomized, double-blind, placebo-controlled, parallel-group study	Men and women aged 50–85 years diagnosed with mild-to-moderate AD as per the 2011 NIA-AA criteria and with 6-month cognitive decline, MMSE (18–26), and with positive amyloid scan	CT1812 300 mg or 100 mg orally compared to placebo for up to 180 days	Number of participants with treatment-related adverse events and serious adverse events over 30 weeks; and changes in brain synaptic density over six months using the SV2A PET ligand 11C-UCB-J
NCT02570997 and NCT03716427 [17,37]	Grundman et al.	2019	USA	A phase 1 clinical trial of the sigma-2 receptor complex allosteric antagonist CT1812, a novel therapeutic candidate for Alzheimer’s disease	Alzheimer’s and Dementia: Translational Research and Clinical Interventions	Phase I	A two-part, randomized, double-blind, placebo-controlled study	Men and women aged between 18–55, or 65–75 years in good health with BMI (19–34 km/m^2^), normal ECG and blood pressure, and non-smokers	Part A: Healthy young subjects with one dose of CT1812 at 10 mg, increased to 30, 90, 180, 450, and 1120 mg across six cohorts; the seventh cohort received a single 90 mg dosage; all dosages were given over 3 days Part B: Healthy young subjects with the same dose once daily for 14 days at 280 mg, followed by 560 and 840 mg in three cohorts; the fourth cohort had healthy elderly subjects aged 65 or above and received 560 mg compared to a placebo daily for two weeks	Safety and tolerability; and plasma pharmacokinetics in parts A (single ascending dose-SAD/food-effect study) and B (multiple ascending dose-MAD)

**Table 2 life-13-00001-t002:** Patient characteristics, safety, and efficacy outcomes.

Author, Year	Sample Size	Age (Years)	Gender	Efficacy	Safety
Izzo et al., 2021 [29]	19	70.2 years (SD = 9.2)	9 F, 10 M	Day 28: Amyloid beta(Aβ) oligomers’ concentration measured via Western blot analysis in CT1812-treated AD patient CSF increases compared to the patient baseline and vs. placebo (*p* = 0.014, *t*-test, n = 3 placebo, 10 CT1812-treated), provided supporting evidence of clinical target engagement; synaptic damage proteins neurogranin B concentration measured via ELISA and synaptotagmin-1 C measured by LC-MS/MS decreased compared to the patient baseline and vs. placebo (*p* = 0.05 covariance, placebo = 5, CT1812 = 11 and *p* = 0.011, placebo = 4, CT1812 = 9), respectively	No adverse events concerning the safety of participants were reported
Hamby, 2020 [36]	23	70 years (SD = 8.8)	11 F, 12 M	Until day 180: the distribution volume ratio (DVR) mean (SE) change from baseline in 300 mg, 100 mg, and placebo groups was −0.043 (0.02), −0.019 (0.02), and 0 (0.02), respectively. Aβ 40 (pg/mL) mean change (SE) over 6 months for 300 mg, 100 mg, and placebo groups were −594 (381.57), 517.7 (445.24), and 222.25 (397.27), respectively. In comparison, for Tau (pg/mL), the 6-month mean (SE) score outcomes were −84.08 (111.96), 121.22 (131.96), and 36.74 (120.7), respectively. Concerning change from the baseline in the cognitive composite (6 ASAS-COG items and 4 NTB items), the 300 mg, 100 mg, and placebo groups had mean (SE) score changes of −0.33 (0.109), −0.11 (0.102) and −0.1 (0.109), respectively. The mean change (SE) for MMSE outcomes in 300 mg, 100 mg, and placebo groups were −2.92 (1.54), −1.26 (1.45), and −0.74 (1.55), respectively.	With high-dose CT1812 (300 mg), mild treatment-emergent adverse events (TEAE) were observed in 85.7% of patients, whereas 4/8 of low-dose (100 mg) patients (50%) reported mild TEAE; with placebo, 4/6 (66.7%) reported mild TEAE; discontinuation was somewhat comparable in both CT1812 (4/15; 26.7%) and placebo (1/6; 16.7%) groups
Grundman et al., 2019 [17]	93; SAD = 54, MAD = 39	SAD Phase: median = 26 years (IQR = 19–55); MAD Phase: young cohort = 28.5 years (IQR = 19–600, elderly cohort = 69 years (IQR = 64–73)	SAD: 16 F, 38 M MAD: 11 F, 28 M	Day 14: CT1812 concentrations in the CSF increased in a dose-dependent manner across two orders of magnitude over two weeks; for 560–840 mg, the average CSF concentrations were at 97–98% receptor occupancy; cognitive scores were similar across the elderly cohort before and after treatment	Treatment-emergent adverse events were reported for 18/42 patients (43%) after single-dose administration of CT1812 and 2/12 patients (17%) after placebo. No deaths or other serious adverse events were reported

**Table 3 life-13-00001-t003:** Characteristics of Ongoing Clinical Trials.

No.	NCT Number	Title	Status	Conditions	Interventions	Outcome Measures	Participants	Phases	Enrollment	Study Type	Study Designs	Other IDs	Primary Completion Date	Locations
1	NCT05531656	A Study to Evaluate the Safety and Efficacy of CT1812 in Early Alzheimer’s Disease	Not yet recruiting	Early Alzheimer’s Disease	CT1812 vs. Placebo	Change from baseline in CDR-SB scale; ADAS-Cog 13; ADCS-ADL; CSF concentrations; Plasma measures of changes; Volumetric MRI including hippocampal and whole brain volume change	M/F, 50–85 years	Phase II	540	Interventional	Randomized, Parallel Assignment, Quadruple (Participant, Care Provider, Investigator, Outcomes Assessor) Marking, Curative	COG0203	Aug-26	NA
2	NCT04735536	Pilot Clinical Study of CT1812 in Mild to Moderate Alzheimer’s Disease Using EEG	Recruiting	Alzheimer Disease	CT1812 vs. Placebo	Measurement of CT1812 plasma concentration ratio	M/F, 50–85 years	Phase II	16	Interventional	Randomized, Crossover Assignment, Quadruple (Participant, Care Provider, Investigator, Outcomes Assessor) Masking, Curative	COG0202	30-Mar-23	Amsterdam, Netherlands
3	NCT03507790	A Study to Evaluate the Safety and Efficacy of CT1812 in Subjects With Mild to Moderate Alzheimer’s Disease.	Recruiting	Mild to Moderate Alzheimer’s Disease	CT1812 vs. Placebo	Number of study participants with treatment-related adverse events and serious adverse events	M/F, 50–85 years	Phase II	144	Interventional	Randomized, Parallel Assignment, Quadruple (Participant, Care Provider, Investigator, Outcomes Assessor) Making, Curative	COG0201	24-Sep-23	United States and Australia

Abbreviations: ADAS-Cog 13: Alzheimer’s disease assessment scale–cognition; ADCS-ADL: activities of daily living scale; CDR: clinical dementia rating scale sum of boxes; CSF: cerebrospinal fluid; MRI: magnetic resonance imaging.

## Data Availability

All data utilized for this study are available publicly and online.

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
