# Peer review of "The Allosteric Antagonist of the Sigma-2 Receptors—Elayta (CT1812) as a Therapeutic Candidate for Mild to Moderate Alzheimer’s Disease: A Scoping Systematic Review"

_life, 2022, doi:10.3390/life13010001_

Round 1

Reviewer 1 Report

The manuscript by Rasheed et al. presents clinical trials on the working of CT1812 (Elayta) on Alzheimer's disease through binding to the Sigma-2 receptor. The subject is interesting, but the manuscript, especially in the Introduction section, contains serious errors which must be corrected.

The issues:

1. Lines 40-47: according to the best knowledge of the reviewer, the mechanism described here is entirely different. The Authors cite [15] and [16] references, and even there, the mechanism is shown differently than in the manuscript.

2. Line 39: "leading to an 'Icelandic' mutation": It looks like a complete misunderstanding of the mechanism. The reviewer is not aware of any literature showing that CT1812 is mutagenic. Even cited [13] and [14] references do not show that. "Icelandic" mutation is a mutation in the Amyloid Precursor Protein (APP, A673T). Once again, according to the reviewer's knowledge, there is no data on CT1812 leading to any mutation.

3. Title and the main text, "Sigma-2 Receptor Complex": CT1812 binds to Sigma-2 Receptor. If the Authors want to write about the "complex" of Sigma-2 Receptor with other proteins, they should explicitly mention the protein partner for the receptor in each case they write it. Otherwise, only the "Sigma-2 Receptor" name should be used. The name "Sigma-2 Receptor Complex" is sometimes used in the literature, but it is incorrect and misleading.

4. Lines 42 and 376: the Authors incorrectly use "ab" (alpha-beta) instead of "Ab" (A-beta).

5. The Sigma-2 Receptor is sometimes called TMEM97, and the Authors use it (e.g., line 345) but never say it is the same protein. It should be clarified.

6. The Authors should explicitly write what were the concerns raised by them in the bias determination.

7. The type of paper should not be assigned as Article, but rather as Review.

Author Response

Reviewer 1 Comments and Author Responses:

The manuscript by Rasheed et al. presents clinical trials on the working of CT1812 (Elayta) on Alzheimer's disease through binding to the Sigma-2 receptor. The subject is interesting, but the manuscript, especially in the Introduction section, contains serious errors which must be corrected. The issues:

Comment 1: 

Lines 40-47: according to the best knowledge of the reviewer, the mechanism described here is entirely different. The Authors cite [15] and [16] references, and even there, the mechanism is shown differently than in the manuscript.

Author Response to Comment 1:

To the reviewer, I invite you to read the updated mechanism as follows:

CT1812 is a highly brain penetrant that works by displacing Aβ oligomers that are bound to neuronal receptors at the synapses. As a lipophilic isoindoline, CT1812 is formed as a fumarate salt that works similarly to the related class of compounds, which have a high specificity and affinity for the sigma-2 receptor complex; this complex is a central regulator of oligomer receptors. The binding of sigma-2 receptors destabilizes the Aβ oligomers binding site, leads to increases in the off-rate of the oligomers, and these are then cleared into the cerebrospinal fluid. Pre-clinical models find that the drugs in the CT family occupy or exceed 80% prevention of downstream synaptotoxicity and restore memory in aged transgenic mouse models of Alzheimer’s disease.

Please read through the following citations: New citations 17-19; these will match the manuscript’s information and are accurate now, to the best of the author’s awareness and clinical knowledge.

Comment 2: 

Line 39: "leading to an 'Icelandic' mutation": It looks like a complete misunderstanding of the mechanism. The reviewer is not aware of any literature showing that CT1812 is mutagenic. Even cited [13] and [14] references do not show that. "Icelandic" mutation is a mutation in the Amyloid Precursor Protein (APP, A673T). Once again, according to the reviewer's knowledge, there is no data on CT1812 leading to any mutation.

Author Response to Comment 1:

While there have been some reports mentioning some interplay and mutation, the authors have decided to omit the sentence entirely to ensure adequate and correct information in our manuscript.

Comment 3: 

Title and the main text, "Sigma-2 Receptor Complex": CT1812 binds to Sigma-2 Receptor. If the Authors want to write about the "complex" of Sigma-2 Receptor with other proteins, they should explicitly mention the protein partner for the receptor in each case they write it. Otherwise, only the "Sigma-2 Receptor" name should be used. The name "Sigma-2 Receptor Complex" is sometimes used in the literature, but it is incorrect and misleading.

Author Response to Comment 3:

Thank you for your much needed insight. You will see the terminology complex removed in writing throughout the text. I have highlighted the words from where I have removed the terminology in yellow.

Comment 4: 

Lines 42 and 376: the Authors incorrectly use "ab" (alpha-beta) instead of "Ab" (A-beta).

Author Response to Comment 4:

Thank you for your much needed insight. I have highlighted the words from where I have updated the terminology in yellow.

Comment 5: 

The Sigma-2 Receptor is sometimes called TMEM97, and the Authors use it (e.g., line 345) but never say it is the same protein. It should be clarified.

Author Response to Comment 5:

Please note that the following sentence has been added: “The authors studied TMEM97, which is also a sigma intracellular receptor 2.” This is stated for clarification.

Comment 6: 

The Authors should explicitly write what were the concerns raised by them in the bias determination.

Author Response to Comment 6:

Please revert to section 3.4. and the limitations section. Various changes have been made, highlighted in yellow for your perusal.

Comment 7: 

The type of paper should not be assigned as Article, but rather as Review.

Author Response to Comment 7:

The article type has been updated.

I thank you for reviewing our manuscript and providing helpful insight into improving outlooks. 

Best Regards,

Dr. Zouina Sarfraz et al.

Reviewer 2 Report

I would like to congratulate the authors on their amazing work. They provided a highly significant research paper since it provides evidence regarding a new therapeutic candidate for one of the most clinically challenging diseases.

The manuscript reports about the use of CT1812 as a Therapeutic Candidate for mild to moderate Alzheimer’s Disease. The authors presented 3 published clinical trials and 3 ongoing clinical trials. I found the manuscript very good and timely done. Nonetheless, there are a number of minor shortcomings:

1-     The title: based on the type of analysis conducted and the depth of the presented data and the discussion, I recommend changing the title to A scoping review rather than a systematic review.

The Allosteric Antagonist of the Sigma-2 Receptor Complex –Elayta (CT1812) as a Therapeutic Candidate for Mild to Moderate Alzheimer’s Disease: A Scoping review

2-     In section 3.4. (Quality Appraisal of the Included Trials), I would recommend adding more details about the identified biases and concerns and try to link this to the validity of the clinical trials findings in the discussion section.

 3-     in the discussion I would recommend adding more about the clinical impact of the findings of the included studies and what these findings could change in practice from the authors perspective.

Author Response

Reviewer 2 Comments and Author Responses:

I would like to congratulate the authors on their amazing work. They provided a highly significant research paper since it provides evidence regarding a new therapeutic candidate for one of the most clinically challenging diseases.

The manuscript reports about the use of CT1812 as a Therapeutic Candidate for mild to moderate Alzheimer’s Disease. The authors presented 3 published clinical trials and 3 ongoing clinical trials. I found the manuscript very good and timely done. Nonetheless, there are a number of minor shortcomings:

Comment 1: 

The title: based on the type of analysis conducted and the depth of the presented data and the discussion, I recommend changing the title to A scoping review rather than a systematic review.

The Allosteric Antagonist of the Sigma-2 Receptor Complex –Elayta (CT1812) as a Therapeutic Candidate for Mild to Moderate Alzheimer’s Disease: A Scoping review

Author Response to Comment 1: 

Thank you for your comment. I have added the term scoping and have renamed it to “Scoping Systematic.”

Comment 2: 

In section 3.4. (Quality Appraisal of the Included Trials), I would recommend adding more details about the identified biases and concerns and try to link this to the validity of the clinical trials findings in the discussion section.

Author Response to Comment 2:

Please revert to section 3.4. and the limitations section. Various changes have been made, highlighted in yellow for your perusal.

Comment 3: 

In the discussion I would recommend adding more about the clinical impact of the findings of the included studies and what these findings could change in practice from the authors perspective.

Author Response to Comment 3:

Please review the new section: 4.2. Implications for Clinical Care

Our review summarizes evidence from existing literature in the area of improving clinical care for Alzheimer’s disease. As a candidate therapy, CT1812 is still under development for age-related degenerative diseases. With currently ongoing trials in this area, as listed in Table 3, we anticipate more robust and concrete evidence solidifying the use of CT1812 for patient populations. Notably, there are currently no approved therapies for dementia with Lewy bodies; this impacts nearly 1.4 million people in the United States alone, as the second most common form of dementia. Individuals that present with motor deficits, cognitive and behavioral changes make it difficult to di-agnose them with any specific type of dementia. With preliminary evidence pointing towards CT1812 effectiveness in reversing deficits in in-vitro models, the various proof-of-concept studies have now progressed towards Phase II of testing. With such approaches, the care for an incurable symptom, dementia, may achieve palpable pro-gress. With prospective placebo-controlled trials, the authors of this study believe that elderly individuals with dementia due to Alzheimer’s disease may have expanded treatment options.

I thank you for reviewing our manuscript and providing helpful insight into improving outlooks. 

Best Regards,

Dr. Zouina Sarfraz et al.

Round 2

Reviewer 1 Report

The Authors made appropriate corrections, as requested.